# I won't listen if I think we're losing our way: How right-wing authoritarianism affects the response to different anti-prejudice messages

**Ayoub Bouguettaya**[1]*, **Matteo Vergani**[2], **Chloe Sainsbury**[1], **Ana-Maria Bliuc**[3]

**1** University of Birmingham, Birmingham, West Midlands, United Kingdom, **2** Deakin University, Melbourne, Victoria, Australia, **3** University of Dundee, Dundee, Nerthergate, United Kingdom

* a.bouguettaya@bham.ac.uk

**Data Availability Statement:** The data underlying the results presented in the study are available from the OSF database (https://osf.io/wxjcd/).

## Abstract

Prejudice reduction messages have been shown to be effective through changing norms. Previous research suggests that Right Wing Authoritarianism (RWA) moderates the reaction to these messages, but it is unclear whether individuals high in RWA are more or less sensitive to prejudice-reduction campaigns. This research used the social identity approach to investigate the role of RWA in moderating the reactions to messages that look to reduce support for prejudicial policies and associated prejudice against an ethnoreligious group (Muslims). Americans (N = 388) were presented with statements on a real, proposed ban on Muslim immigration into the US from an in-group member (i.e., an American freight worker who disapproves of the Muslim ban), outgroup member (an Iraqi refugee who is in favour if the Muslim ban), or both, or control message. Those high in RWA showed consistently high levels of prejudice against Muslims in all conditions, but those low in RWA showed lower prejudice when presented with the anti-prejudice message from an in-group member (compared to control). This suggests that anti-prejudice messages primarily affect those with low RWA, clarifying that RWA likely leads to resistance to anti-prejudice messages regardless of the source. Future research aiming to reduce prejudice should examine how messages can be tailored to reduce prejudice in those with high RWA.

## Introduction

Decades of research have examined how to reduce prejudice effectively in the general population [1]. Most research has focused on the value of intergroup contact [2] or the importance of modifying intergroup norms [3], or both [4]. However, research has also shown that some important factors beyond the control of experimenters moderates the impact of these prejudice reduction strategies. One factor that which has been demonstrated repeatedly to be important is right-wing authoritarianism (RWA); this personality style appears moderates the reaction to these anti-prejudice strategies. Some research shows that higher RWA results in greater resistance to some anti-prejudice strategies [5,6], while other research shows that higher RWA actually results in greater drops in prejudice upon exposure to anti-prejudice

**Funding:** The funding for this study was provided by Deakin University's Alfred Deakin Fellowship, provided to the second author, who also provided a salary for the second author. We have acknowledged the role of funders in the manuscript, and wish to acknowledge the funders did not have a role in drafting, analysing, or writing this manuscript.

**Competing interests:** The authors have declared that no competing interests exis

strategies, such as intergroup contact [7,8]. Furthermore, some research has also indicated that context generally changes how any message designed to change norms is perceived [9–11], and other research has shown that those high in RWA are also more sensitive to contextual framing when it comes to their interpretations of intergroup norms [12]. It is therefore plausible that the efficacy of prejudice reduction messages may entirely depend on the context frame of the message, and the level of RWA of the individual receiving that message (which may interact).

In this experimental study, we sought to examine how the effectiveness of an anti-prejudice message could be affected by context frame (i.e., contrasting statements) and RWA. By using the social identity approach, this study also attempted to present a novel prejudice reduction strategy. This approach relied on the use of normative messages, and enhancing the power of those messages by manipulating the frame of reference (comparative contrast). Our research focused on omnipresent rhetoric around a ban by then President Donald Trump in 2017 and 2018, proposing "a total and complete shutdown of Muslims entering the United States," and his subsequent executive orders mandating a ban on immigration from people in multiple Muslim majority countries. This order was prejudiced, and support for such a ban acts as an externally valid measure of prejudice.

## Understanding prejudice: Personal and social identity approaches

Broadly, prejudice is defined as an antipathy, largely based on stereotypes about other groups (and its members; [13,14]. Prejudice is largely a reflection of intergroup relations [15], suggesting that how groups see each other is a fundamental part of prejudicial beliefs. Social identity approaches posit that prejudice is a possible consequence of a desire to maintain a positive *social identity* (or sense of oneself as a group member [16–19]. This approach posits that individuals have an inherent desire to maintain positive distinctiveness by increasing their group's collective self-esteem above others [20]. Prejudice arises because self-categorisation as a group member leads to less motivation to differentiate out-group members (leading to stereotypes), and out of a desire to maintain positive distinctiveness, these stereotypes are usually negative [16,21]. Discrimination, in the form of in-group bias, usually follows to ensure these intergroup relations are maintained [22].

Although prejudice is commonly considered from the perspective of group identification, probably the single strongest factor in understanding whether or not a person will have prejudicial beliefs is one's personality style, not one's social groups [8,23]. It has been established that personal belief systems appear to be the strongest predictor of prejudice [24]. The main belief systems that appears to predict prejudice is Right Wing Authoritarianism, or RWA [24,25], along with social dominance orientation (see Dual process model in [26]). RWA is a personality style, characterised by conventionalism, submission to authorities, and hostility/aggression to those who defy authority or convention [27]. Those high in RWA are far more likely to have prejudicial attitudes toward homosexual individuals [28], support state-based racial discrimination [29], and engage in aggression against ethnoreligious minorities [30]. Therefore, in order to reduce prejudice, prejudice reduction strategies would conceivably aim to reduce prejudice in those high in RWA.

It appears to be that RWA is a belief system underpinned by a particular definition of what it means to be a group member. From the social identity approach, this means that RWA is in fact, a form of normative fit—where an individual's knowledge-based expectations determine their perception of what it means to be categorised as a "group" [31]. This affects the boundaries of who is an "in-group" member, and who is an "outgroup" member. High RWA means stricter boundaries around norm defiance, and anyone who defies or blurs those boundaries is

not considered a group member. For example, while an American with low RWA may consider a Muslim American as an American, a high RWA American would consider this individual to have defied norms around religion, and would not consider that person as an American in context.

As would be predicted by such an understanding of RWA, research has shown that RWA differentially predicts prejudice toward different groups, depending on whether they are seen to defy in-group norms or not. For example, RWA predicts prejudice toward homosexuals [28] but not "gay men and lesbian women" [32]. This is because the former group defies in-group norms (and therefore, do not fit with one's in-group), while the latter are not seen as a threat [33]. This suggests that understanding RWA from an SIA may lead to better anti-prejudice strategies.

## Changing prejudice: Integrating RWA and social identity approaches for better strategies in reducing prejudice against threatening groups

Generally, there is conflicting evidence in regards to how specifically RWA in general moderates the response to anti-prejudice strategies against threatening groups; it has been suggested most approaches fail or make prejudice worse in those high in RWA [5,6]. One key branch of anti-prejudice reduction strategies, intergroup contact, suggests the opposite. From a social identity approach, positive intergroup contact reduces prejudice and changes intergroup beliefs [4]. Exactly how is still a topic of considerable debate [4], but some research suggests that positive intergroup contact changes intergroup norms (what makes them, *them* across different categories). Research suggests that those high on RWA actually are more sensitive to prejudice reduction strategies that rely on positive intergroup contact, and show corresponding drops in prejudice [7,8].

However, there are two main problems with strategies to reducing prejudice on the basis of intergroup contact. First, most individuals high on ideological intolerance (like RWA) will avoid positive intergroup contact with threatening groups, and instead will seek environmental contexts that only feed prejudice [34,35]. This means that these strategies, although efficacious, are unlikely to reach those who need it most. Furthermore, more recent evidence suggests that these contact approaches do not generalise to outside that context; one pre-registered study suggested that positive intergroup contact between Muslims and Christians in a soccer team did not improve relations outside of soccer, because the sense of threat was not alleviated in the intervention. [36] Second, most prejudice reduction strategies actually take the form of public messages dictating norms [37–39], as intergroup contact is expensive and largely difficult to set up in a natural way. Research has also suggested that those high on RWA are more sensitive to contextual and frame of reference effects [32]. Together, these problems suggest that the most common form of prejudice reduction strategy—normative messaging—must consider context as those most likely to be prejudiced (high RWA) are more likely to consider the context of these messages.

## The present study: Reducing prejudice in those high in RWA via the social identity approach

The social identity approach helps explain how there is a discontinuity between individual and group behaviour [40]; the attitudes and behaviours that individual display is a mix of both personal and social identity. There is a significant amount of evidence showing individual differences in prejudicial beliefs (RWA) are moderated by social identity effects of salience and identity content [25,41,42]. In a study examining prejudice against a variety of groups in French society, prejudice against Arabs significantly correlated with RWA only under certain

conditions [12]. This study split 179 French students into separate conditions, which either emphasised self-categorisation as a member of a group with no competitive norm primed (e.g., being part of an ethnic group formed on the basis of sharing cultural traditions), or a competitive group (for example a sports team, or political party). Results showed that RWA did predict prejudice when group identity was emphasised, but not when social norms regarding intergroup competition were made salient [12]. This suggests that social identity salience and content can shift how personal variables relate to prejudice, and therefore social context is critical.

This study will attempt to use the social identity approach to change the intergroup beliefs against the most disliked minority in the United States (Muslims; [43]). It is believed that anti-Muslim prejudice in the United States can be partially accounted for by a intergroup normative belief that Muslims are, inherently un-American due to being violent [44]. To reduce this belief, it may be needed to change the norms within American society. A number of experimental studies have indicated that a normative belief of "who belongs" can be altered by manipulating a frame of reference [9–11]. In other words, a person previously considered to be an "out-group" member can become part of the in-group when the reference frame is manipulated, which then may reduce prejudice. Furthermore, research has shown that a normative message from an in-group member can become more influential when paired with an outgroup member [11]. In this study, it was found that when moderate feminists were presented with an extreme feminist message alone, they did not find it influential. However, when the same extreme message was presented with an anti-feminist message, participants accepted the extreme message and found it influential. Furthermore, recent experimental evidence suggests that if an outgroup member makes a statement that aligns with a person's in-group (rendering them atypical within their outgroup), then the frame of reference may also change, resulting in reduced prejudice [45]. Therefore, what "we" are can be affected by what "we" are not, and messages from atypical outgroup members that fit with our in-group beliefs can change what we believe "we" are not. In this way, it is possible to change the norms of inclusion in a desired direction, by presenting contrasting statements from in-group members and out-group members.

Our article presents a social identity approach as an exploration for how messages can prejudice against Muslims in an American sample through understanding how RWA may affect the response to these messages. Specifically, our overall aim was to understand how RWA may moderate the reaction to anti-prejudice messages against Muslims. We predicted:

Hypothesis 1 (H1): presenting a statement from an out-group member endorsing an anti-Muslim norm for the American identity (though support for a Muslim ban) will reduce prejudice and reduce support for associated policies, as will a message from an in-group member endorsing a pro-inclusion norm for Muslims (through the latter's rejection for a Muslim ban).

Hypothesis 2 (H2): Presenting these messages together will result in a significantly lower amount of prejudice and support prejudicial policies than either message alone.

Hypothesis 3 (H3): RWA will moderate this relationship between contextual effects and prejudice. Those high in RWA will show significantly weaker prejudice reduction effects in every manipulation condition, except when both statements are presented (where they will show much greater reductions in prejudice, as the heightened comparative context sensitivity will override intergroup beliefs).

## Method

These methods were pre-registered on OSF (link redacted for publication).

## Measures

**Right wing authoritarianism.** This very short form of the RWA scale [46] presented participants with six statements, and asked them to indicate their level of agreement with these statements on a scale from 1-Very strongly disagree to 9-very strongly agree. This scale has been used to detect how much an individual believes in submitting to authority, and how hostile they are to individuals who do not adhere to the societal conventions. The mean score was used.

**Prejudice scale.** This prejudice scale [47] presented participants with a series of statements against a particular minority (modified here to discuss Muslims). While the original scale assessed five subscales, this study focused on one subscale, asking participants to indicate their agreement or disagreement on six statements (scale from 1-strongly disagree to 7-strongly agree). This scale was the Threat and Rejection Blatant Scale, (e.g., "Americans and Muslims can never be really comfortable with each other, even if they are close friends,"). This subscale assessed the extent to which they felt that Muslims were a threat or danger to Americans. We chose this scale as the dominant stereotype against Muslims in the United States suggests Muslims are dangerous [44]. Higher values indicated greater prejudice. However, for inclusion, we also assessed a merged scale which examined blatant threat and subtle threat together.

**Participants and procedure.** In November 2018, Americans who voted in the 2016 Election from Amazon's Mechanical Turk (N = 388, $M_{age}$ = 36.11 years old, 60.1% Male) were invited to participate in a short, online survey titled "Understanding social cohesion and intergroup attitudes". Participants were first presented with a plain language statement, demonstrating ethical approval by the host university, and asked to confirm they were American and consent to the parameters in the plain language statements. Participants were then asked to insert their MTurk IDs, and some basic demographic information (i.e., age, year of birth). In the next step, as part of our manipulation examining the effect and context of messages, participants were randomly assigned to one of four conditions: in-group message only, out-group message only, both messages (to see effect of context), and control message.

They were then instructed that there were to be presented with a statement from people on a policy that they would need to think about, and press next when they comprehend. Participants were warned they would be asked about the details of the message later on (a manipulation check). In the control condition, we presented a hypothetical scenario where there were debates on raising the standards for federal driving licences, making it more difficult. In this context, participants were told that a radio station sought to get opinions on this, and interviewed a number of people on this issue. On the next page, a male freight worker was asked about this increase, and he expressed that he, as an American, was unsure about it, appealing to the law (Declaration of Independence on freedom), and appealing to current states (current Americans with licences). In all other conditions, participants were told about how the Supreme Court recently allowed the so called "Muslim ban", blocking people from primarily Muslim countries (regardless of reason of entry). The same radio station sought to get opinions, so they interviewed people. In the in-group condition, the same American freight worker was asked about the Muslim ban. He expressed that he, as an American, felt it was un-American, appealing to the law (First Amendment), and appealing to current states (current Muslims who fit what it meant to be law-abiding hard-working Americans). In the out-group condition, an Iraqi freight worker was asked about the ban from the same American radio station. This Iraqi expressed that, as a non-American, it was exactly what America is, making an appeal to the illegitimacy of their laws (calling the constitution filled will lies), and appealing to current states (stating that all they care about is war and how being Muslim is seen to be incompatible

with being American). In the last condition (dual message condition), both the in-group and outgroup message on the ban were presented together.

Participants were then asked about their previous awareness of the ban, their support for the ban, and whether they were aware of the recent Iraq wars. After this, the survey was identical to Study 1 (all the same measures, except for strength of identification sub-scales dropped from analyses in Study 1). Participants were then thanked, provided the opportunity to respond to the survey via written feedback, and compensated $1 USD for their time. This project was approved by the host university's ethics committee.

**Manipulation check.** We had three manipulation check questions. These manipulation checks asked participants about their comprehension about the messages, asking who the message source was, their views, and who the interviewer was. If anyone failed the attention check, they were presented with the messages again, with a timer that would not let them proceed until one minute had passed. We had 16 participants who failed at least two attention checks (not included in the N = 388). Half of these participants also showed a lack of consistency in their responses in scales, with most completing the scales in a near impossible timeframe (i.e., in a matter of seconds). One of these participants stated that they disliked the survey and deliberately manipulated their responses to make it unlikely for us to find anything. These participants were removed from the analyses. We also asked participants about their knowledge of the Iraq war. 100% of participants were aware of the Iraq war.

## Results

### Descriptive statistics

Means, standard deviations, alphas, and correlations are reported in Table 1. We found that generally RWA correlated with both threat prejudice and support for the prejudicial policy of a Muslim ban ($p < .001$). Support for ban was notably low across all conditions ($M = 2.62$, $SD = 2.02$). Threat prejudice against Muslims was less varied (Blatant threat $M = 2.66$, $SD = 1.25$). Anonymised data is available on OSF.

### Inferential statistics

To test the effect of the different messages against H1 and H2, a one-way multivariate analysis of variance was conducted to examine whether our manipulation was successful at reducing

**Table 1. Descriptive statistics and correlations.**

| | Mean (SD) | Alphas | Age | Trump Warmth | Support of ban | Conservatism | RWAaver | Strengthties | SDOaver | PrjThreatBlant | PrejThreTrad |
|---|---|---|---|---|---|---|---|---|---|---|---|
| Age | 36.11 (10.70) | - | 1 | | | | | | | | |
| Trump Warmth | 32.29 (38.38) | - | -.148** | 1 | | | | | | | |
| Support of Ban | 2.62 (2.02) | - | -.071 | .691** | 1 | | | | | | |
| Conservatism | 3.53(1.86) | - | -.191** | .776** | .614** | 1 | | | | | |
| RWAaver | 4.14(1.38) | .560 | -.061 | .569** | .532** | .605** | 1 | | | | |
| Strengthties | 5.18(1.31) | .900 | -.049 | .415** | .302** | .429** | .377** | 1 | | | |
| PrjThreatBlant | 2.66(1.25) | .860 | -.124* | .659** | .764** | .617** | .576** | .262** | .424** | 1 | |
| PrejThreTrad | 3.12(1.39) | .924 | -.135** | .712** | .800** | .705** | .593** | .33** | .453** | .940** | 1 |

Note: N = 388.

* p < .05

** p < .01.

**Table 2. Means, standard errors, and 95% Cis.**

| Condition | Ban support | | Threat prejudice | |
|---|---|---|---|---|
| | *M (SE)* | *95% CI* | *M (SE)* | *95% CI* |
| Control (N = 99) | 2.71 (.20) | 2.31–3.11 | 2.74 (.13) | 2.49–2.98 |
| Ingroup only (N = 100) | 2.45 (.20) | 2.05–2.85 | 2.53 (.13) | 2.29–2.78 |
| Outgroup only (N = 92) | 2.73 (.21) | 2.31–3.14 | 2.71 (.13) | 2.45–2.97 |
| Outgroup-ingroup (N = 97) | 2.61 (.21) | 2.21–3.01 | 2.65 (.13) | 2.40–2.90 |

threat prejudice or support for the Muslim ban. The multi-variance effect for condition was insignificant, Wilks' Lambada $F(6, 766) = .28$ p = 0.95 $\eta^2 = .002$. The follow-up univariate tests found that the difference between conditions was insignificant for both outcome variables (p < .947). Table 2 shows the means, standard errors, and 95% Confidence intervals for scores within each condition. Interestingly, we found the message from the in-group member appeared to reduce Trump warmth compared to the control message ($t(197) = 2.876$, p < .01, Cohen's d = .41) This suggests that the manipulation was not broadly successful at reducing blatant threat prejudice nor support for the ban, indicating that H1 and H2 were not supported.

To test H3 that higher RWA results in significantly weaker prejudice reduction effects except when both statements are presented, we first tested whether higher RWA generally lead to higher threat prejudice. A median split for RWA was conducted to enable comparison between high RWA and low RWA groups, and we found that those high in RWA generally did have higher prejudice regardless of condition ($t(386) = 11.27$, p < .001), as well as higher support for the Muslim ban ($t(386) = 10.11$, p < .001). The one-way multivariate analysis of variance was repeated using the RWA median split, and the results showed that contrary to H3, those high in RWA showed no change across conditions on threat perceptions ($F(3,176) = .357$, p = .784), and those low in RWA also showed no change across conditions ($F(3,204) = 1.874$, p = .135). However, follow-up tests showed that participants with low RWA did show lower prejudice when presented with an in-group member's rejection of the Muslim ban, compared to control statements ($t(106) = 2.692$, p = .008), but other statements had no significant effect.

Although the statements generally had no significant effect between conditions and between participants with low vs high RWA, we believed it was worth examining reactivity to messages between RWA types. Those high in RWA showed higher prejudice in the control condition (N = 48, M = 3.19, SD = 1.128) than those low in RWA (N = 51, M = 2.30, SD = 1.166). Fig 1 illustrates this effect (t = 3.827, DF = 96.92, p = .000. CI of difference: 0.456–1.341), and those high in RWA showed higher prejudice in the ingroup member condition (N = 43, M = 3.43, SD = 0.977) than those low in RWA (N = 57, M = 1.85, SD = .746) (CI of difference 1.186–1.906). A regression with intercept showed that the main effect of RWA was significant (t = 89.253, p = .000), the main effect of condition (control vs ingroup member only) was insignificant (t = 1.48, p = .306), and the interaction between RWA and condition was significant ($t(1) = 11.087$, p = .001). This significant interaction suggests that the uneven effect holds; RWA moderates the response to anti-prejudicial ingroup messages in that those high in RWA will ignore these messages, while those with low RWA will attend to these messages.

## Discussion

The aim of our study was to understand how we might be able to reduce prejudice against Muslims, and if personal factors might moderate a response to an anti-prejudice message

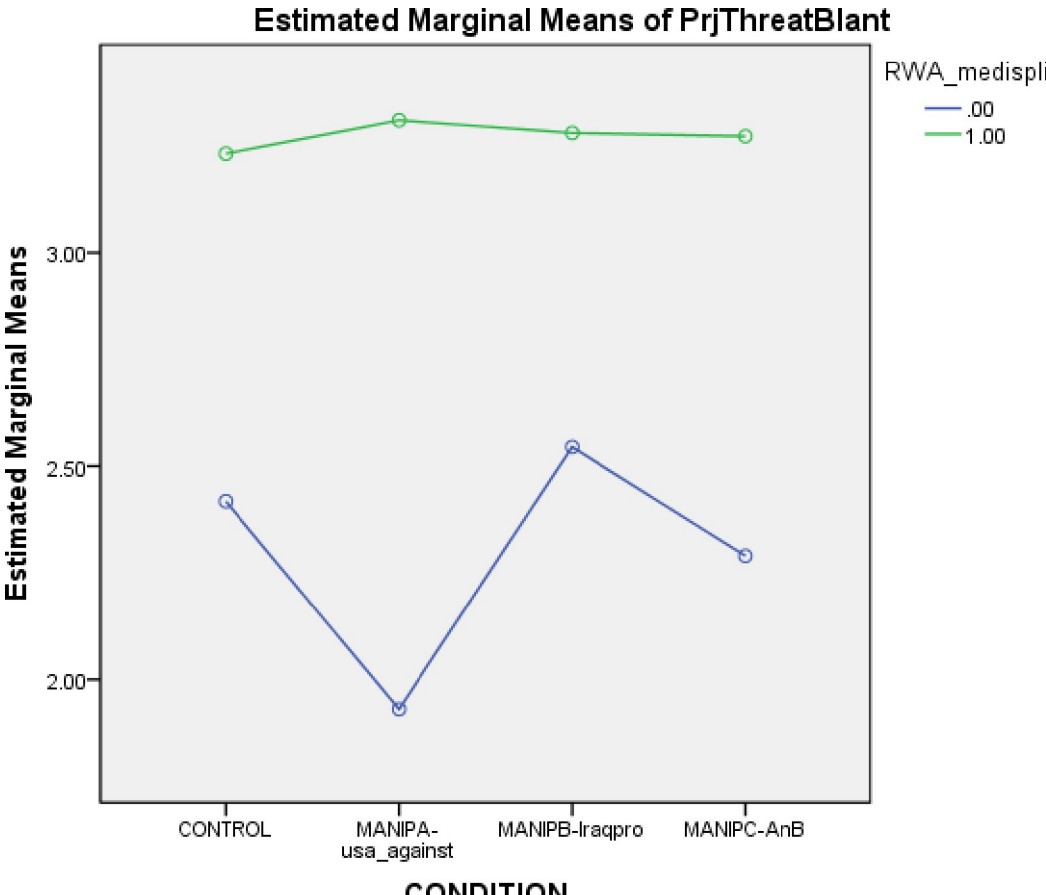

**Fig 1. Condition split by RWA for blatant threat prejudice, against the four conditions of control, message A (against prejudice) message B (Iraq for prejudice) and message A&B combined.**

based on context. Contrary to what was expected, our manipulations had little to no effect overall between conditions. Therefore, H1 and H2 were unsupported. Surprisingly, in no condition did the dual message condition (i.e., the condition in which we had a contrasting statement) have any effect in reducing prejudice beyond the in-group message, suggesting that the contrast was largely ineffective. Some support was found for H3; as expected, those high in RWA did not change in prejudice or support for the ban regardless of condition (although they also did not respond to the dual message either, which was contrary to the hypothesis). Meanwhile, those low in RWA showed reduced prejudice in response to an in-group member expressing disapproval of the ban.

Researchers have begun to develop more complex models of how intergroup contexts and personal beliefs (such as RWA) interact to predict intergroup beliefs, such as prejudice [28,30,41,48]. The value of tailoring anti-prejudice messages for specific audiences, however, has largely been ignored, and the importance of how these messages can be weakened or strengthened by context has similarly not been considered in contemporary literature. Alone, the ingroup member's message appeared to reduce prejudice for those low in RWA compared to the control condition, but not for those high in RWA.

In this study, higher RWA predicted prejudice overall. Similarly, RWA predicted one's response to an anti-prejudice message; those high in RWA did not show any changes in prejudicial beliefs across all conditions, while those low in RWA showed lower prejudicial beliefs

when presented with a message from an in-group member denouncing a prejudicial policy relative to the control condition. Overall, these findings suggest that "traditional" anti-prejudice normative messages from in-group members appear to only work on individuals low on RWA.

Past research has suggested that prejudice is a function of intergroup relationships [16,21] but more recent research has suggested that personal traits dictate one's response to prevailing intergroup relations [48]. Theoretically, social identity approaches suggest an individual's perception of the norms of their group matter in regard to how they react to social contexts. Furthermore, research in this field has also suggested that personal perceptions of one's group, such as RWA, dictate how they see "distance" between groups, and this interacts with the social context broadly [25,32,48]. Very recent research has suggested that those high in RWA tend to increase in prejudice over time [49]. Our research hypothesized that while a personal belief can dictate in-group beliefs and prejudice, we also hypothesized these beliefs mattered regarding how individuals would react to specific messages and their contexts. Our study found mixed evidence against these assertions, as the manipulations generally did not result in lower levels of prejudice generally, and the interaction between the personal variables (RWA) and the messages did not always occur in the direction expected.

While we did not see reverse effects (i.e., increased prejudice as a reaction), previous research has found that prejudice-reduction messages can backfire and produce the opposite of the intended effect; Legault, et al ([38] demonstrated that these messages can actually increase prejudice, depending on the type of messaging used. 103 non-Black students in Toronto were randomly allocated into conditions comparing the effects of anti-prejudice brochures. The brochures which emphasised reducing prejudice as an autonomous decision which benefits society were shown to cause a decrease in prejudice, while brochures explaining reducing prejudice as a social requirement to comply with pressures of political correctness were shown to cause an increase in prejudice, demonstrating that prejudice-reduction messages require careful consideration to avoid this reverse effect. Furthermore, Álvarez-Benjumea and Winter [50] demonstrate the importance of context framing in prejudice reduction; 274 German residents were randomly allocated to conditions where they viewed only positive, only neutral, or a mix of positive, neutral and negative messages about refugees after terrorist attacks in Germany. When exposed to negative messages, there was an increase in prejudiced hate speech against refugees post-terrorist attack, but this prejudice did not increase (and even slightly decreased) when the only context provided was positive or neutral messages. Therefore, the context (or the frame) in which prejudice-reduction messages are viewed and presented is crucial to ensuring the intended effect of decreasing prejudice.

The findings of this study suggest the value of integrating personal factors with social factors in understanding and reducing prejudice. As such, we argue that prejudice should be understood as an individual's response to a prevailing social context, and their personal beliefs (of the norms within their social identity) relate to that context (see [51] on identity navigation). This is why RWA was the key variable that dictated one's prejudicial beliefs, and reaction to an in-group member denouncing a prejudicial policy. We suggest that RWA is largely an internalised group norm (i.e., a social belief of "who we are" inspired by one's immediate social group) dictating normative fit, as it leads to more rigid rules, including deference to authority. Americans with high RWA are likely to consider Muslims a threat no matter the context, as the past leadership (Donald Trump) has clearly dictated Muslims are un-American, claiming Islam "hates us" [52], an existential threat through his statement on the ban [53], and refusing to rule out the possibility of a Muslim register [54]. Trump also suggested that those who disagree with him generally as un-American [55]. As such, there is a possibility that those high in RWA would not attend to the in-group member's message because they discounted the source as "not truly one of *us as Americans*" via the principle of normative fit (i.e., rejecting a group

member who shows atypical traits as a group member; [31], thereby causing the source to lose their ability to influence. Although it was believed that manipulating the context to make it so the in-group member was a prototypical American by contrast, it appears that this was ineffective broadly, and RWA resulted in a lack of engagement from a peer, regardless of *any* message given by this otherwise influential person. Social identity approaches have long argued that influence is a function of similarity to the group in the form of prototypicality [56], but it may be that the mere act of saying that the Muslim ban is un-American, those high in RWA would find that message discounting a threat to their identity. Any given anti-prejudice intervention against racism or immigration may need to be more directed toward the very idea of threat instead to redirect threat fears in order to be more palatable against the perceived ingroup norms amongst those high in RWA.

The findings of this study are believed to be applicable to suggesting against certain mechanisms for reducing prejudice against threatening groups. Further research should attempt to investigate what factors can reduce prejudice in those with high RWA; as highlighted previously there is considerable debate on what messages work on those with high RWA. Prejudice reduction messages likely work on those low in RWA, but they are less likely to show prejudice regardless [30]. Instead, RWA itself should be targeted broadly to reduce prejudice. As it has been described that RWA dictates what norms are attended to, other social identity approaches may be needed. For example, making other identities salient may serve to change one's attitudes toward an outgroup [57], or as RWA causes greater deference to authority ([27,58,59], changing the message source from a leader (e.g., George W Bush stating that Muslims are our friends and Americans amongst high RWA Republicans) may be the other ways prejudice can be reduced.

Overall, our findings suggest that understanding and attempting to reduce prejudice requires an integrative understanding of social and personal identity, and how they interact. The social identity approach is perhaps the best theoretical explanation of intergroup behaviour, and as such, this paper suggests that integrating personal explanations of prejudice with social explanations can lead to a meta-theoretical paradigm to understand, predict, and reduce prejudice through public messages. While there is recent research showing how social contexts and personal variables interact to reduce prejudice and support for prejudicial policies ([48], a meta-theoretical paradigm has not been established, despite evidence suggesting that the social identity approach is likely the best approach. This lack of cohesion within psychological explanations of human and social behaviour has been argued to be a key driver in the replication crisis in psychology [60], as there are numerous micro-level theoretical approaches that have been relegated to explaining findings, rather than driving revisions of meta-theoretical paradigms of human behaviour. Through our findings, we argue that integration of personal and social explanations of prejudice may aid in creating more replicable and robust research in understanding the reasons behind prejudice, and how to best reduce it through an examination of the social identity approach. The main novelty to this work is that it demonstrates that contrasts likely do not work, and the findings suggest that the assessment of the impact of these interventions needs to be more fine-tuned.

## Limitations

This research had a few key limitations in the manipulations and in the measurement tools. In measuring the support for the Muslim ban, there were two key issues. First, the so-called "Muslim ban" has had a number of changes in the past few years. We asked a question that fit the original wording of the ban by Donald Trump, but the last version of the ban was directed at certain Muslim majority countries. Therefore, participants may have responded to the ban

message differently. Some of the qualitative responses stated this was the case, while some tried to respond according to the question, others stated they disliked the current version of the ban because it did not go far enough. Second, the measure may not have been sensitive enough. It was one question, and there may have been nuance lost in the context of the study. A series of questions asking about different levels of ban would have likely been more sensitive to different levels of support, like questions asking about a ban on all adult Muslim men who have completed military service in a foreign country, or a ban on young Muslim children. These issues with this scale may have introduced unexplained variance into our study, and future research should address this by collapsing the question into prejudicial scales that assess migration support as a function of prejudice.

Another potential limitation was that the manipulation used may also have been too weak or inappropriate for this study. There is a possibility that the out-group member was simply not "out-group" enough. We chose this out-group member because the natural contrast to American is a person from a different nationality, and that group member would have been affected by the ban, but it may have been better to either choose a rival identity (e.g., a Canadian), or to make the contrast much more salient (e.g., describe the individual as extremely un-American). This is critical to the theoretical paradigm used; the social identity approach suggests that the difference between one's own group and an out-group is assessed on a degree of perceived similarity (via the meta-contrast ratio; [61]). Intergroup beliefs only start to affect attitudes and behaviour when this difference leads to a significant gap between the observer (i.e., our participant) and the target (the outgroup member; [9]). Other research (N = 1034) even suggests that RWA is actually associated with stronger acquisition of more positive views toward prejudiced groups [62]; it may have possible that our ingroup member was also just not positive enough. Future research should investigate how to manipulate the contrast enough to elicit these intergroup effects; some research has already indicated that this ratio is affected by personal variables, which further emphasizes the need for follow-up research in prejudice reduction messages. This particular intervention may need more refinement against the target group, potentially borrowing research from anti-stigma-based messages as well.

## Conclusion

Prejudice is a serious, worldwide problem in Western society. While this study's findings are directed against Muslims, the research here is likely to be applicable to other groups that are perceived to be "dangerous", but may not have applicability against prejudice against groups that are a symbolic threat [33]. This research suggests that the social identity approach, conjoined with a personality approach, to prejudice is probably the best theoretical approach to explain and reduce prejudice against Muslims and other "dangerous groups". This paper shows that prevailing approaches [2,38] to reduce prejudice through public messaging from an in-group member may work, but as per previous research [48], only those low in RWA will respond in the desired direction by showing lower prejudice as a reaction. This research also adds to the social identity literature on reducing prejudice; it demonstrates the importance how an individual's navigation of their social identities matters just as much as social contexts. This research also goes some way to address the otherwise inconsistent findings and views on how RWA impacts the response to anti-prejudice messages; context (or frame of reference) likely does not affect antiprejudice beliefs in those high in RWA, reflecting a broad resistance to these types of messages. Future research using this approach must consider how intergroup contrasts occur as a function of context. Understanding how prejudice is related to personal and social variables is critical to ensure a more peaceful, harmonious society for good governance, and learning how reduce prejudice may change the lives of millions for the better.

## Acknowledgments

We would like to thank the anonymous peer reviewers for this manuscript for their work. We appreciate your time.

## Author Contributions

**Conceptualization:** Ayoub Bouguettaya, Matteo Vergani, Ana-Maria Bliuc.

**Data curation:** Ayoub Bouguettaya, Matteo Vergani.

**Formal analysis:** Ayoub Bouguettaya, Matteo Vergani, Ana-Maria Bliuc.

**Funding acquisition:** Matteo Vergani.

**Investigation:** Ayoub Bouguettaya, Matteo Vergani.

**Methodology:** Ayoub Bouguettaya, Matteo Vergani.

**Project administration:** Ayoub Bouguettaya, Ana-Maria Bliuc.

**Resources:** Ayoub Bouguettaya.

**Software:** Ayoub Bouguettaya.

**Supervision:** Ana-Maria Bliuc.

**Validation:** Ayoub Bouguettaya, Matteo Vergani, Chloe Sainsbury.

**Visualization:** Ayoub Bouguettaya, Chloe Sainsbury.

**Writing – original draft:** Ayoub Bouguettaya.

**Writing – review & editing:** Ayoub Bouguettaya, Matteo Vergani, Chloe Sainsbury, Ana-Maria Bliuc.

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
