## [Decision Letter · Decision Letter 0]

23 May 2022

PONE-D-22-00364I’ll won’t listen if I think we’re losing our way: How right-wing authoritarianism affects the response to different anti-prejudice messagesPLOS ONE

Dear Dr. Bouguettaya,

Thank you for submitting your manuscript to PLOS ONE. After careful consideration, we feel that it has merit but does not fully meet PLOS ONE’s publication criteria as it currently stands. Therefore, we invite you to submit a revised version of the manuscript that addresses the points raised during the review process.

The manuscript examines interesting and timely questions, but further work is required before it is ready for publication. A revised version of the manuscript should address the methodological and theoretical considerations raised by both reviewers (noted in their reviews), including justifying the analysis decisions and expanding reviews of related research. Importantly, the noted preregistration plan should be made available for examination in the next round of reviews.

We look forward to receiving your revised manuscript.

Kind regards,

Dr. Alexandra Paxton

Academic Editor

PLOS ONE

Journal Requirements:

Reviewers' comments:

Reviewer's Responses to Questions

**Comments to the Author**

1. Is the manuscript technically sound, and do the data support the conclusions?

Reviewer #1: Partly

Reviewer #2: Yes

2. Has the statistical analysis been performed appropriately and rigorously? 

Reviewer #1: Yes

Reviewer #2: Yes

3. Have the authors made all data underlying the findings in their manuscript fully available?

Reviewer #1: Yes

Reviewer #2: Yes

4. Is the manuscript presented in an intelligible fashion and written in standard English?

Reviewer #1: Yes

Reviewer #2: No

5. Review Comments to the Author

Reviewer #1: The authors address an important topic: the degree to which individuals who are high in right wing authoritarianism (RWA) are more or less sensitive to prejudice reduction campaigns. This speaks to a growing body of literature seeking to understand what kinds of prejudice-reduction interventions may be most effective in diverse contexts. That said, below I raise several concerns the authors should address before publication:

1) The authors sample size seems quite small, with no group containing more than 100 subjects. To what degree are the authors null findings a true null or just a consequence of the study being underpowered?

2) Recent research suggests that as of 2018, as many as 25% of respondents participating on MTurk were found to be fraudulent. What checks have the authors done to ensure that their respondents are genuinely participating?

3) How much variation is there in RWA in the sample? Does splitting the sample at the median appropriately capture differences between high and low RWA participants?

Reviewer #2: The authors investigate an important topic and provide sufficient justification based on prior literature. There is only partial support for their various hypotheses which limit the contributions of this paper. I would suggest the authors address thee things in a revision:

1) The authors assume in-group membership for the in-group stimulus member but have no manipulation check to ensure he was perceived as an in-group member. This is an important limitation and should be explicit.

2) Related to number one above, it is overly simplistic to assume one chosen in-group and one chosen out-group member will be representative of either. Egon Brunswick argued that we should use a sample of stimulus for the same reasons we use samples of participants. It is a huge assumption to assume that the single in-group member is typical or representative of participant in-group members.

3) The authors should review more literature on "anti-Muslim attitudes" and "authoritarianism." There are significant works not addressed by the authors.

6. PLOS authors have the option to publish the peer review history of their article (what does this mean?). If published, this will include your full peer review and any attached files.

Reviewer #1: No

Reviewer #2: No

---

## [Author Response · Author response to Decision Letter 0]

9 Jun 2022

Dear Editor,

Thank you for the opportunity to revise this manuscript. Below, we address each of the reviewer’s comments and editor’s requests.

Editor’s requests:

Thank you for these suggestions. We have addressed this. 

Thank you for these suggestions. We have amended the cover letter accordingly.

We have added this as a section, labelled ethics. 

We have added the URL in the cover letter.

https://osf.io/wxjcd/

We have checked and found all studies cited.

Reviewer 1:

Reviewer #1: The authors address an important topic: the degree to which individuals who are high in right wing authoritarianism (RWA) are more or less sensitive to prejudice reduction campaigns. This speaks to a growing body of literature seeking to understand what kinds of prejudice-reduction interventions may be most effective in diverse contexts. That said, below I raise several concerns the authors should address before publication:

1) The authors sample size seems quite small, with no group containing more than 100 subjects. To what degree are the authors null findings a true null or just a consequence of the study being underpowered?

We did a power analysis prior to running the study, but due to word count, we removed this. We have restored this in the manuscript, lines 217-222.

To provide a little more detail than in the manuscript, we calculated the effect size by looking at the relevant t-test in the manuscript (Dru, 2007): 

“The regression coefficient for RWA in the group values condition was marginally divergent from the one calculated for the competitive condition (t(118)= 1.89, p=.06). “ That converts to a cohen’s D of .35, or effect size of F= 0.175. 

We now say in the revised manuscript: 

In November 2018, Americans who voted in the 2016 Election from Amazon’s Mechanical Turk (N=388 (retained), Mage= 36.11 years old, 60.1% Male) were invited to participate in a short, online survey titled “Understanding social cohesion and intergroup attitudes”. We had aimed to recruit at least 396 participants, based our sample size calculation on Dru’s 2007 study (12), which ran an anti-prejudice experiment while examining RWA. They found an effect size of F= 0.175 (df=188), for a similar intervention. Our power analysis for an ANOVA using the same effect size (a=.05, power=.80) suggested we needed 360 participants, but we aimed to over recruit by 10% in order to remove those who failed attention checks.

2) Recent research suggests that as of 2018, as many as 25% of respondents participating on MTurk were found to be fraudulent. What checks have the authors done to ensure that their respondents are genuinely participating?

We had multiple attention checks and manipulation checks. Here, we have clarified the content of one of the attention checks. 

We now revised the manuscript as follows:

Manipulation and attention check 

We had three manipulation/attention check questions. These manipulation checks asked participants about their comprehension about the messages, asking who the message source was, their views, and who the interviewer was. If anyone failed the attention check, they were presented with the messages again, with a timer that would not let them proceed until one minute had passed. We had 16 participants who failed at least two attention checks (not included in the N=388). Half of these participants also showed a lack of consistency in their responses in scales, with most completing the scales in a near impossible timeframe (i.e., in a matter of seconds). One quarter of these participants also failed a simple attention check where they were told to select a specific response to the question (e.g., “please select the option that shows you do not accept a given statement”). One of these participants stated that they disliked the survey and deliberately manipulated their responses to make it unlikely for us to find anything. These participants were removed from the analyses. We also asked participants about their knowledge of the Iraq war. 100% of participants were aware of the Iraq war.

3) How much variation is there in RWA in the sample? Does splitting the sample at the median appropriately capture differences between high and low RWA participants?

There was slightly more variation in the sample at the high level of RWA, while low levels of RWA had considerably less variance (i.e., it was positively skewed). The median was 4, with the lowest possible score of 1 and the highest possible score of 8. This suggests that those low in RWA were fairly similar, while those high in RWA were had a bigger range. We have provided a bit more detail in the manuscript about this (ln 282-286)

A median split for RWA (Median=4.00) was conducted to enable comparison between high RWA and low RWA groups; this fit the data as there was clustering at the bottom half of the scale (suggesting similarity), with more extremity at the top end. This was reflected in higher variance in the top half of scores for RWA (SD= .934) than in the lower half (SD=.622)

Reviewer 2:

 The authors investigate an important topic and provide sufficient justification based on prior literature. There is only partial support for their various hypotheses which limit the contributions of this paper. I would suggest the authors address thee things in a revision:

1) The authors assume in-group membership for the in-group stimulus member but have no manipulation check to ensure he was perceived as an in-group member. This is an important limitation and should be explicit.

2) Related to number one above, it is overly simplistic to assume one chosen in-group and one chosen out-group member will be representative of either. Egon Brunswick argued that we should use a sample of stimulus for the same reasons we use samples of participants. It is a huge assumption to assume that the single in-group member is typical or representative of participant in-group members.

We have added a paragraph to make these limitations clearer (ln 450-464): 

Extending on this same limitation, there is a possibility that the ingroup member was not seen as “ingroup enough”. We did not have a manipulation check to assess whether or not participants saw ingroup member as being relatively similar. We assumed that the ingroup member would be influential due to shared identity, but this was not assessed. We also assumed that the mere presence of the outgroup member would shift that ingroup member to become more representative, but this was an assumption based on theoretical and previous evidence, rather than being tested in this study. This limitation could be addressed in future research in two ways. First, future research should pilot multiple ingroup prototypes that cover a range of ingroup norms, and attempt to align specific prototypes against the norms espoused by participants. For example, a rural American may respond better to a prejudice intervention by someone espousing values more common in rural areas. Second, future research should assess how “ingroup” a prototype is, before presenting a message from that member. Together, these may lead to more powerful interventions that are more custom fit, thereby potentially improving their impacts on those traditionally resistant to anti-prejudice messages.

3) The authors should review more literature on "anti-Muslim attitudes" and "authoritarianism." There are significant works not addressed by the authors.

We have now discussed more literature on each of the points. For example, in relation to anti-Muslim attitudes we have also reviewed:

Dunwoody PT, McFarland SG. Support for anti‐Muslim policies: The role of political traits and threat perception. Political psychology. 2018 Feb;39(1):89-106.

Uenal, F., Bergh, R., Sidanius, J., Zick, A., Kimel, S. and Kunst, J.R., 2021. The nature of Islamophobia: A test of a tripartite view in five countries. Personality and Social Psychology Bulletin, 47(2), pp.275-292.

For Authoritarianism, we included a detailed discussion of:

Hanson K, O'Dwyer E, Lyons E. The national divide: A social representations approach to US political identity. European Journal of Social Psychology. 2021 Jun;51(4-5):833-46.

---

## [Decision Letter · Decision Letter 1]

30 Aug 2022

PONE-D-22-00364R1I’ll won’t listen if I think we’re losing our way: How right-wing authoritarianism affects the response to different anti-prejudice messagesPLOS ONE

Dear Dr. Bouguettaya,

Thank you for submitting your manuscript to PLOS ONE. After careful consideration, we feel that it has merit but does not fully meet PLOS ONE’s publication criteria as it currently stands. Therefore, we invite you to submit a revised version of the manuscript that addresses the points raised during the review process.

One of the original reviewers was unable to review the revision, and it took some time before we were able to secure a new reviewer. The continuing reviewer (Reviewer 2) remains positive about the quality of the writing but raises a question about the novelty and impact of the work. The new reviewer (Reviewer 3) points out some concerns about scope and audience. If you and your coauthors decide to submit a revision, you should expand the review of the literature and discussion to more firmly situate the work and, in so doing, appropriately outline its novel contributions and impacts (along with providing some additional detail and revisions requested by Reviewer 3).

We look forward to receiving your revised manuscript.

Kind regards,

Alexandra Paxton

Academic Editor

PLOS ONE

Journal Requirements:

Additional Editor Comments (if provided):

Reviewers' comments:

Reviewer's Responses to Questions

**Comments to the Author**

1. If the authors have adequately addressed your comments raised in a previous round of review and you feel that this manuscript is now acceptable for publication, you may indicate that here to bypass the “Comments to the Author” section, enter your conflict of interest statement in the “Confidential to Editor” section, and submit your "Accept" recommendation.

Reviewer #2: (No Response)

Reviewer #3: (No Response)

2. Is the manuscript technically sound, and do the data support the conclusions?

Reviewer #2: Yes

Reviewer #3: Partly

3. Has the statistical analysis been performed appropriately and rigorously? 

Reviewer #2: Yes

Reviewer #3: Yes

4. Have the authors made all data underlying the findings in their manuscript fully available?

Reviewer #2: Yes

Reviewer #3: Yes

5. Is the manuscript presented in an intelligible fashion and written in standard English?

Reviewer #2: Yes

Reviewer #3: Yes

6. Review Comments to the Author

Reviewer #2: The article is well written and concise. Unfortunately, it adds little value to the published research.

Reviewer #3: Title- I suggest correct the grammar ie I won’t listen

Abstract

Americans may know what is meant by ‘the Muslim ban’ but I had to read all the way to p19 for an explanation of this. Please write for an international audience.

The conclusion that RWA likely leads to resistance to all antiprejudice messages regardless of the source is overstated. These findings can’t be generalised to e.g. messaging delivered by peers or influential figures. Further, the experimental conditions are so far from the methods used by successful antiprejudice campaigns (such as Time to Change in England and Wales) I am not sure how useful the findings based on them are. Instead I think researchers should consider working with those who deliver such campaigns to evaluate them in a rigorous way; the delivery is much more sophisticated than what academics can deliver in an experiment, but academic evaluation is often lacking for social marketing campaign, so this would be a win win.

Methods

I would not know what to make of a question about Americans and Muslims friendships- I would get stuck at the thought ‘what about all the Muslim Americans’? Were such questions subjected to cognitive testing first?

7. PLOS authors have the option to publish the peer review history of their article (what does this mean?). If published, this will include your full peer review and any attached files.

Reviewer #2: No

Reviewer #3: **Yes: **Professor Claire Henderson

---

## [Author Response · Author response to Decision Letter 1]

25 Sep 2022

We would like to thank all three reviewers for their feedback to adjust this manuscript. Below, we address all comments. 

Reviewer #2: 

The article is well written and concise. Unfortunately, it adds little value to the published research.

We appreciate these comments on the quality of the manuscript. We hope that our amendments, in response to reviewer 3, add to the published research in a constructive way. 

Reviewer #3: 

Title- I suggest correct the grammar ie I won’t listen

We apologise for this; this was a direct quote from our pilot participants. We have amended the title. 

Americans may know what is meant by ‘the Muslim ban’ but I had to read all the way to p19 for an explanation of this. Please write for an international audience.

Thank you for this comment. We have added a statement of clarification on both the abstract, and on line 61.

The conclusion that RWA likely leads to resistance to all antiprejudice messages regardless of the source is overstated.

We have removed the word “all” as there was a potential clarity issue here. What we meant was that the peer source likely doesn’t make much of a difference with people high in RWA. 

These findings can’t be generalised to e.g. messaging delivered by peers or influential figures. 

Further, the experimental conditions are so far from the methods used by successful antiprejudice campaigns (such as Time to Change in England and Wales) I am not sure how useful the findings based on them are. 

We believe a bit of refinement is needed here. Here, we have added a bit of discussion on generalisation on lines 110, 112, 123,129, 383- pointing out two things: the social influence value, and the impact of anti-prejudice interventions against the topic:

-Social influence

From a social identity approach, the distinction between leader or peer based social influence is blurred. The group is the source of the influence, with prototypicality of a group member mattering more than anything else (Steffens et al, 2018). Therefore, how similar one is to the perceived group’s values, needs, desires, and state is where leaders and peers get their influence from, interacting with how people see the group. People are influential because they share group membership and embody what it means to be a group member. Our method of attempting to influence people via these prototypical Americans is against this approach. To clarify this, we have expanded a discussion on line 383 onward. 

Steffens, N. K., Haslam, S. A., Jetten, J., & Mols, F. (2018). Our followers are lions, theirs are sheep: How social identity shapes theories about followership and social influence. Political Psychology, 39(1), 23-42.

-Different versions of prejudice

It is believed that prejudice toward things like mental health, disability, ageism, etc, likely function on a different level than things like immigrants, Muslims, Mexicans in the US, Indian/Pakistan views, Balkan prejudice etc. The former is more about symbolic threat- they’re a threat to how we live our lives, and the latter is a threat to our very existence (Rios et al, 2018). Therefore, we would have to adjust accordingly. 

Anti-prejudice strategies, therefore, need to be tailored against what the threat is. This is why a contact intervention that seeks to build understanding between Muslims and Christians in Iraq fails to decrease prejudice between them outside of that specific context (Mousa et al, 2020), while a contact intervention that seeks to reduce ageism (Christian et al, 2014), or prejudice against people with mental health conditions likely would work (Morgan et al, 2018). 

We’ve made some small adjustments on lines 110, 112 , 123 and 129 to address this to some extent. We have also mentioned this in limitations as well, line 451, 458. 

Christian, J., Turner, R., Holt, N., Larkin, M., & Cotler, J. H. (2014). Does intergenerational contact reduce ageism: When and how contact interventions actually work?. Journal of Arts and Humanities, 3(1), 1-15.

Morgan, A. J., Reavley, N. J., Ross, A., San Too, L., & Jorm, A. F. (2018). Interventions to reduce stigma towards people with severe mental illness: Systematic review and meta-analysis. Journal of psychiatric research, 103, 120-133.

Rios, K., Sosa, N., & Osborn, H. (2018). An experimental approach to intergroup threat theory: Manipulations, moderators, and consequences of realistic vs. symbolic threat. European Review of Social Psychology, 29(1), 212-255.

Instead I think researchers should consider working with those who deliver such campaigns to evaluate them in a rigorous way; the delivery is much more sophisticated than what academics can deliver in an experiment, but academic evaluation is often lacking for social marketing campaign, so this would be a win win.

We agree that would be ideal but here, we see this as more of a theoretical test. Does the contrast message shift people’s prejudice? Here, the answer is no, especially as those who likely needed the anti-prejudice most didn’t respond across all conditions. This is a consistent problem in anti-prejudice research, as highlighted in the paper.

On a more pessimistic view, the problem is very difficult to solve broadly. It is the lead author’s personal belief from working in government and in academia, and academic opinion, that most campaigns only shift people who have positive view to be more positive, while those who hold these views do not change. In order to address prejudice, schooling, shifts in society dynamics, and changes of why threats exist is far better. No message-based marketing campaign is likely to reduce prejudice in a meaningful way without additional work across more than one domain. 

Methods

I would not know what to make of a question about Americans and Muslims friendships- I would get stuck at the thought ‘what about all the Muslim Americans’? Were such questions subjected to cognitive testing first?

The questions were largely taken from established scales, with very strong validity and reliability. While such a method might be a good idea for future research, it is largely out of scope for this proof of concept, as we were using an established scale previously used. This allows direct comparison with the other studies using this scale. 

Pettigrew TF, Meertens RW. Subtle and blatant prejudice in Western Europe. European journal of social psychology. 1995;25(1):57-75.

---

## [Decision Letter · Decision Letter 2]

4 Jan 2023

I won’t listen if I think we’re losing our way: How right-wing authoritarianism affects the response to different anti-prejudice messages

PONE-D-22-00364R2

Dear Dr. Bouguettaya,

We’re pleased to inform you that your manuscript has been judged scientifically suitable for publication and will be formally accepted for publication once it meets all outstanding technical requirements.

Kind regards,

Shrisha Rao, Ph.D.

Academic Editor

PLOS ONE

Additional Editor Comments (optional):

Reviewers' comments:

Reviewer's Responses to Questions

**Comments to the Author**

1. If the authors have adequately addressed your comments raised in a previous round of review and you feel that this manuscript is now acceptable for publication, you may indicate that here to bypass the “Comments to the Author” section, enter your conflict of interest statement in the “Confidential to Editor” section, and submit your "Accept" recommendation.

Reviewer #3: All comments have been addressed

2. Is the manuscript technically sound, and do the data support the conclusions?

Reviewer #3: Yes

3. Has the statistical analysis been performed appropriately and rigorously? 

Reviewer #3: Yes

4. Have the authors made all data underlying the findings in their manuscript fully available?

Reviewer #3: Yes

5. Is the manuscript presented in an intelligible fashion and written in standard English?

Reviewer #3: Yes

6. Review Comments to the Author

Reviewer #3: no further comments

7. PLOS authors have the option to publish the peer review history of their article (what does this mean?). If published, this will include your full peer review and any attached files.

Reviewer #3: **Yes: **Claire Henderson

---

## [Editor Report · Acceptance letter]

6 Jan 2023

PONE-D-22-00364R2 

I won’t listen if I think we’re losing our way: How right-wing authoritarianism affects the response to different anti-prejudice messages 

Dear Dr. Bouguettaya:

I'm pleased to inform you that your manuscript has been deemed suitable for publication in PLOS ONE. Congratulations! Your manuscript is now with our production department. 

Kind regards, 

on behalf of

Dr. Shrisha Rao 

Academic Editor

PLOS ONE